# Determinants of College Students' Actual Use of AI-Based Systems: An Extension of the Technology Acceptance Model

**Kang Li**

School of Economics and Management, Zhejiang University of Water Resources and Electric Power, Hangzhou 310018, China; lik@zjweu.edu.cn

**Abstract:** Acceptance of, behavioral intention towards, and actual use of AI-based systems or programs has been a topic of growing interest in the field of education. A considerable number of studies has been conducted to investigate the driving factors affecting users'/students' intentions regarding certain technology or programs. However, few studies have been performed to understand college students' actual use of AI-based systems. Moreover, the mediating effect of students' learning motivation was seldom considered. Therefore, the present study was conducted to explain factors contributing to college students' actual use of AI-based systems, as well as to examine the role of their learning motivations. As a result, perceived usefulness and perceived ease of use of AI-based systems positively impacted students' attitude, behavioral intentions, and their final, actual use of AI-based systems, while college students' attitude towards AI-based systems showed an insignificant impact on students' learning motivations of achieving their goals and subjective norms. Collectively, the findings of the present study could enrich the knowledge of the technology acceptance model (TAM) and the application of the TAM to explain students' behavior in terms of the adoption of AI-based systems.

**Keywords:** technology acceptance; AI-based systems; learning motivation; college students; PLS-SEM

## 1. Introduction

With the development and extensive application of the Internet, wireless networks, sensing technologies, and mobile technologies, innovative changes in teaching and learning have been witnessed. This especially considers the restrictions on mass gatherings conducted by governments due to the outbreak of COVID-19 in late 2019. The academic system was among the first sectors to be seriously impacted, and all classes were disallowed based on the social distancing requirements used as a preventive measure. Naturally, the application of learning with the use of the Internet and other online technologies has extensively spread and been adopted by diverse levels of schools, such as elementary schools, universities, etc., and an increasing number of learning processes are becoming online-based and are incorporating technologies related to automating assessments or helping instructors in the process of their teaching [1–4].

Artificial intelligence (AI) and its relevant applications in various fields has been widely explored and investigated across the world in previous decades, and, among these fields, a particularly emerging topic is the application of AI in students' learning; therefore, various related systems have been developed to facilitate and improve the effectiveness of teaching and the learning of students [4,5]. For instance, a study conducted by Cui, Xue, and Thai indicated that students achieved better performance using the Yixue Squirrel AI adaptive learning system in comparison to both traditional classroom instruction provided by expert teachers and other adaptive learning platforms [6]. Lee, Hwang, and Chen examined the application of AI-based chatbots in the review process of public health courses, and explored whether or not they could improve students' academic performance, self-efficacy, learning attitude, motivation, etc. [7].

Additionally, the technology acceptance model (TAM) was proposed by Davis (1989) based on the theory of reasoned action (TRA), and it was utilized to explain the relationship between students' computer system technologies, behavioral intentions (BI), and definite behavior of technology use (also known as actual use (AU) of certain technology) [8–10]. Later, this research model was extensively adopted by many studies to explain and unveil a user's attitude (Att), BI, or AU of new technology-related programs, systems, or apps, for example, mobile apps to learn language [11], K-MOOCs [12], self-service kiosks in tourism and hospitality [13], applications of Zoom in language courses [14], etc.

It was illustrated that educational platforms and applications are more closely aligned with learners' needs and knowledge, making the educational process more efficient. Thus, AI has great potential in higher education [15]. Therefore, the objectives of this study were to: construct an integrative model to understand the determinants of college students' AU of AI-based systems by combining TAM and students' learning motivations; and examine the relationship among students' technology acceptance, Att, learning motivations, BI, and AU of AI-based systems. Correspondingly, implications for future research directions such as the application of TAM or extended TAM in the understanding of students' adoption of new technology, and the development and influence of AI-based systems in students' study behaviors, could be discussed and established.

Collectively, this study addresses the following research questions: (1) What is the correlation among college students' perceived ease of use (PEOU), perceived usefulness (PU), Att, learning motivation, BI, and AU towards AI-based systems? (2) What are the key factors in determining college students' AU of AI-based systems? (3) What implications could be extracted contributing to the development and improvement of the application of AI-based systems among college students?

## 2. Literature Review

Based on the theory of reasoned action (TRA), developed by Fishbein and Ajzen (1975), Davis (1989), firstly, proposed the technology acceptance model (TAM) to explain the association between students' acceptance of computer systems and their technologies, BI, and definite behavior of technology use [8–10]. According to the theory of technology acceptance, two personal beliefs, which are PU and PEOU, respectively, could predict users' Att towards using a specific technology. Furthermore, the Att itself could impact a user's BI towards a particular technology, which, in turn, could predict the actual system's use of users [9,11]. It has been concluded by previous studies that the application of TAM in educational technology acceptance has proved its effectiveness in explaining students' new learning motivations, behaviors, or performances in terms of different technologies, such as mobile technology, computer-based communication technology, social media, MOOCS courses, etc. [16–18]. Since its invention, many external variables have been added to TAM in order to better explain and predict the acceptance of, and intention to use, information technology systems [17].

### 2.1. Perceived Ease of Use and Perceived Usefulness

PEOU is defined as "the degree to which a person believes that using a particular information system would enhance his/her job performance" [9]. In educational settings, it could be interpreted as the tendency of students or learners to use or not use an application or specific technology based on the extent to which they believe it will enhance their study performance [10,14]. PU is defined as "the degree to which an individual believes that using a particular system is free of physical and mental effort" [9]. PEOU and PU are the two central constructs of TAM, and have been extensively researched to determine their influence on customers' Att and BI towards new technological adoption [19,20]. In the basic TAM, PU directly affects consumers' intention towards technological adoption, while PEOU directly and indirectly (through PU) affects consumers' intentions towards adoption [21].

Previous studies have shown that users perceiving a new technology as useful or easy to use are more likely to adopt it; also, a positive effect of PEOU on the PU of a new technology was verified [17,22]. Therefore, this leads to the following hypothesis:

**Hypothesis 1 (H1):** *Students' perceived ease of use of AI-based systems positively affects their perceived usefulness of AI-based systems.*

### 2.2. Attitude towards AI-Based Systems

Though various definitions are available, scholars have been unable to reach a valid definition of Att. However, Att has generally been referred to as a positive or negative evaluation of people, objects, events, activities, ideas, or environment [13]. Numerous studies have successfully utilized and replicated TAM to predict users' acceptance of novel technologies and systems and demonstrated that PU and PEOU largely determine user Att toward a specific technology, while Att and PU significantly affect BI to use the technology [23].

Specifically, it was suggested that the application of AI-based chatbots in the review process of public health courses could improve students' academic performance, self-efficacy, learning attitude and motivation, i.e., AI-based systems could improve students' learning attitude and motivation in their learning process [7]. Moreover, for language learning, there are positive results regarding perceptions concerning the integration of conversational AI chatbots, especially in relation to PEOU and Att [24]. Therefore, based on the above discussion, the hypotheses between PU, PEUO, Att and BI were developed as:

**Hypothesis 2 (H2):** *Students' perceived ease of use of AI-based systems positively affects their attitude towards AI-based systems.*

**Hypothesis 3 (H3):** *Students' perceived usefulness of AI-based systems positively affects their attitude towards AI-based systems.*

**Hypothesis 4 (H4):** *Students' perceived usefulness of AI-based systems could positively affect their behavioral intentions towards AI-based systems.*

### 2.3. Learning Motivation

Koff and Mullis regarded learning motivation as the students' choice of specific learning activity and the efforts devoted to such activities [25]. Therefore, learning motivation could be defined as the sum of the incentives that positively force the choice of a specific behavior or purpose [26]. Additionally, students' learning behavior could be motivated by students' intrinsic interests, desire to achieve specific goals, or teachers/parents' extrinsic rewards or recommendations [27,28]. Hence, the learning motivation in the present study could be represented by students' learning interest [10,27,29–31], achieving goals [27] and subjective norm [8,9]. Subjective norm (SN) refers to "the person's perception that most people who are important to him or her think he or she should or should not perform the behavior in question and it is related to the importance of social influences on acceptance that affects individual behavior [8,9]. Generally, it is considered a part of the social influence variable and signifies the perceived social pressure to carry out or avoid carrying out a behavior [32]. This concept was adopted as one of the learning motivational factors because, based on the content of TPB, the subjective norm could be the predictive and motivational variable impacting users/consumers' intention [8,33].

In the process of technology-based self-directed learning, students' technological learning motivation is reflected in their mastery and familiarity of technical skills, as is consistent with the study which revealed that technological learning motivation significantly influences students' intention to use online learning websites [10,34]. Thus, according to the above discussion, the hypotheses among PU, PEOU, students' attitude, and aspects of learning motivation could be developed as follows:

**Hypothesis 5a (H5a):** *Students' perceived usefulness of AI-based systems positively affects their learning interest in learning motivation.*

**Hypothesis 5b (H5b):** *Students' perceived usefulness of AI-based systems positively affects their achievement of goals of learning motivation.*

**Hypothesis 5c (H5c):** *Students' perceived usefulness of AI-based systems positively affects their subjective norm of learning motivation.*

**Hypothesis 6a (H6a):** *Students' perceived ease of use of AI-based systems positively affects their learning interest in learning motivation.*

**Hypothesis 6b (H6b):** *Students' perceived ease of use of AI-based systems positively affects their achievement of goals of learning motivation.*

**Hypothesis 6c (H6c):** *Students' perceived ease of use of AI-based systems positively affects their subjective norm of learning motivation.*

**Hypothesis 7a (H7a):** *Students' attitude towards AI-based systems positively affects their learning interest in learning motivation.*

**Hypothesis 7b (H7b):** *Students' attitude towards AI-based systems positively affects their achievement of goals of learning motivation.*

**Hypothesis 7c (H7c):** *Students' attitude towards AI-based systems positively affects their subjective norm of learning motivation.*

*2.4. Behavioral Intention*

Generally, BI refers to the degree to which a person has formulated conscious plans to perform or not perform some specified future behavior [35]. Therefore, regarding AI-based systems, BI means the intent of students to employ AI-based systems and involves persistent use from the present to the future [15,16,36]. Prior studies have indicated that learning motivation (e.g., learning interest, achieving goals and subjective norm) of learners or students could directly and significantly influence their BI. For instance, the study by Shroff and Keyes suggested that students' perceived interest positively impacts their BI [37], and Lee examined that subjective norm could positively affect users' intention to online games [38], etc. Hence, this leads to the correlation between students' learning motivation and BI as follows:

**Hypothesis 8a (H8a):** *Students' learning interest of learning motivation could positively affect their behavioral intention towards AI-based systems.*

**Hypothesis 8b (H8b):** *Students' achievement of goals of learning motivation could positively affect their behavioral intention towards AI-based systems.*

**Hypothesis 8c (H8c):** *Students' subjective norm of learning motivation could positively affect their behavioral intention towards AI-based systems.*

*2.5. Actual Use of AI-Based Systems*

Based on the prior literature, BI is likely to be correlated with actual usage of certain technology, while the variables of PEOU and PU of TAM are less likely to be correlated with actual usage [39]. Therefore, based on the above discussion, the hypotheses among students' learning motivation, BI towards AI-based systems and their AU of AI-based systems were established as:

**Hypothesis 9a (H9a):** *Students' learning interest in learning motivation could positively affect their actual use of AI-based systems.*

**Hypothesis 9b (H9b):** *Students' achievement of goals of learning motivation could positively affect their actual use of AI-based systems.*

**Hypothesis 9c (H9c):** *Students' subjective norm of learning motivation could positively affect their actual use of AI-based systems.*

**Hypothesis 10 (H10):** *Students' behavioral intention towards AI-based systems positively affect their actual use of AI-based systems.*

Collectively, the study investigated the relationship among students' PEOU, PU of AI-based systems, Att, learning motivation, BI, and AU. The conceptual model is represented in Figure 1.

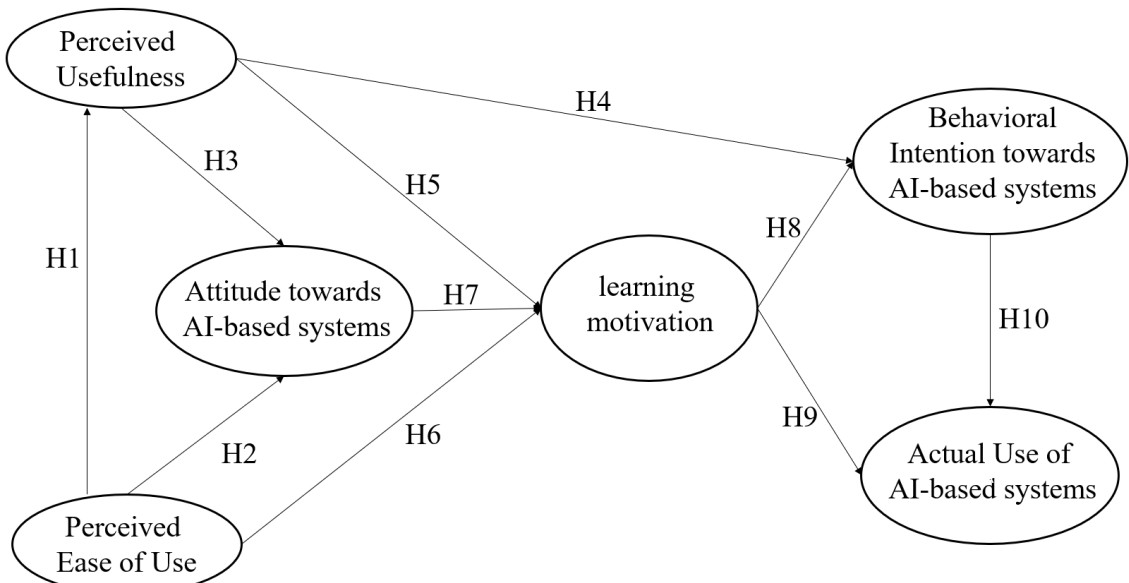

**Figure 1.** Conceptual model.

### 3. Methodology

*3.1. Data Collection*

The data collection was carried out during the spring semester of 2022 by distributing self-administrated surveys among college students. Online surveys through a convenience sampling method were utilized for data collection. Additionally, Wenjuanxing.com, which is a platform quite popular in China providing professional online questionnaire survey, voting, testing and comments, was applied to establish the survey and a unique URL link and QR code were generated to enable participants' access to the questionnaire [40].

The questionnaire survey was opened from 10 May to 20 May until there were no new responses procured; the collected data were downloaded directly from Wenjuanxing.com in the format of a Microsoft excel sheet. At the very beginning of the questionnaire survey, the explanation of AI-based systems has been presented: these are systems or programs built based on the technology of AI. Meanwhile, several examples have been provided for participants' understanding, such as adaptive/personalized learning (Polaris AI Tutor, Youdao Speak), scenario education based on virtual reality (VR) and augmented reality (immersive vr education), and so forth. Moreover, a screening question to ask whether the participant has used AI-based systems for learning was applied. The following questions in the questionnaire continued to be filled out if the respondents answered yes. Otherwise, appreciation would be expressed and the questionnaire survey would be terminated.

Initially, 314 surveys were collected. After deleting invalid questionnaires which were missing values, rated all items with same values, or had been completed within an excessively short time (such as less than 100 s), a total of 279 usable questionnaires were evaluated, providing a valid response rate of 88.9%.

*3.2. Survey Instrument*

A questionnaire was developed based on a thorough literature review and some revisions have been made based on the real situation. There are two major parts of the questionnaire. The first part of the survey asked about students' socio-demographic information (gender, age, major and level of education) and other questions pertinent to AI-based systems for their learning, such as types of AI-based systems they used, the frequency with which they used these systems, etc. The second part consists of diverse questions to evaluate students' PEOU, PU, Att, learning motivation (learning interest, achieving goals and subjective norm), BI and AU of AI-based systems. A Likert 5-point scale was applied to evaluate to what extent the respondents agree with the statements provided in the questionnaire, varying from 1, representing mostly disagree, to 5, standing for mostly agree [41]. The items used in each construct and their relative references were synthesized in Table 1.

**Table 1.** Summarization of survey instruments.

| Constructs | Items | References |
|---|---|---|
| Perceived ease of use (PEOU) (five items) | PEOU1: It is easy for me to learn to use AI-based systems. PEOU2: Leaning by AI-based systems is easy for me. PEOU3: It is easy for me to become proficient in using AI-based systems. PEOU4: My interaction with AI-based systems is easy for me to understand. PEOU5: Overall, I think that AI-based systems are easy to use. | [9,10,42] |
| Perceived usefulness (PU) (five items) | PU1: Using AI-based systems helps me to learn more efficiently. PU2: Studying with AI-based systems would improve my study efficiency. PU3: AI-based systems are advantageous for my learning. PU4: AI-based systems make my study easier without limitation of location and time. PU5: Overall, I think that AI-based systems are useful in my daily study. | [9,10,42] |
| Attitude towards AI-based systems (Att) (four items) | Att1: I like using AI-based systems for my study. Att2: I feel good about using AI-based systems. Att3: Using AI-based systems is an attractive way for me to study. Att4: Overall, my attitude towards AI-based systems is positive. | [8,10,42] |
| Learning motivation-Learning interest (MLI) (five items) | MLI1: I become more interested in my study by using AI-based systems. MLI2: I wish I could improve my study outcome by using AI-based systems. MLI3: I wish I could improve my study outcome by using AI-based systems. MLI4: I believe my interest to learn could be enhanced by using AI-based systems. MLI5: I believe I could be more interested in my study by using AI-based systems. | [10,29,31] |

**Table 1.** *Cont.*

| Constructs | Items | References |
|---|---|---|
| Learning motivation-Achieving goals (MAG) (four items) | MAG1: Learning with AI-based systems is important for me to achieve my goals.<br>MAG2: Learning with AI-based systems is important because I will be able to acquire relative knowledge that I need for my future career and income.<br>MAG3: Learning with AI-based systems is important because it could help me save time and money.<br>MAG4: Learning with AI-based systems is efficient since I could access it anytime and anywhere. | [10,29,30] |
| Learning motivation-Subjective norm (MSN) (six items) | MSN1: My parents encourage me to learn with AI-based systems as much as I can.<br>MSN2: My major teachers support me to learn with AI-based systems.<br>MSN3: My teachers feel it will be instrumental for me to learn with AI-based systems.<br>MSN4: My friends think it will be helpful to use AI-based systems for my study.<br>MSN5: My friends suggest me to learn with AI-based systems.<br>MSN6: Overall, people wo I value encourage me to learn with AI-based systems. | [10,28–31] |
| Behavioral intention towards AI-based systems (BI) (four items) | BI1: I intended to adopt AI-based systems for my study.<br>BI2: I would use AI-based systems if I was given the opportunity.<br>BI3: I think AI-based systems are useful for my study.<br>BI4: The likelihood that I would recommend AI-based systems to my friends/classmates is high. | [10,30,42] |
| Actual use of AI-based systems (AU) (four items) | AU1: I think that AI-based systems help me improve my study performance.<br>AU2: I think that the design of AI-based systems helps me improve my learning performance.<br>AU3: I believe that AI-based systems help me cooperate well with others.<br>AU4: I believe that AI-based systems help me enhance my learning efficiency. | [3,9,14,42] |

*3.3. Data Analysis*

Data analysis of the study was performed with SPSS 23.0 and SmartPLS 3.2.7. SPSS 23.0 was utilized to calculate the skewness and kurtosis values of all items to examine whether the collected data were evenly distributed [43]. Furthermore, a partial least square-structural equation model (PLS-SEM) approach was employed to examine the measurement model and structural model by using SmartPLS 3.2.7. The measurement model (also known as outer model) refers to the relation between the constructs and their indicators, while the structural model refers to the association between the latent constructs themselves [9]. PLS-SEM method is variance-based SEM, which is suitable to analyze relatively small samples. Additionally, the present research is an extension of the existing structural theory (i.e., TAM) and the structural model is, to some extent, quite complex (including a relatively great number of constructs and indicators); thus, PLS-SEM was adopted for analyzing the proposed research model in the present study [44].

## 4. Results
*4.1. Socio-Demographic Characteristics*

A total of 279 survey responses were used for final data analysis. A total of 142 (50.9%) were male students and 137 (49.1%) were female students. In terms of respondents' age, it

was in the range of 17–25 years and the average age is 20.62 years. In total, 185 (66.3%) samples are undergraduates and 94 (33.7%) respondents are graduates. As for the demographic factors of specialization, 48 (17.2%) respondents majored in social science, 96 of the samples come from engineering backgrounds, 34.4%, and 135 (48.4%) respondents have a science and technology background. In terms of the using frequency, most respondents (168) used AI-based systems for learning at least once per day (58.4%), followed by twice to five times a week (72, 25.8%). A total of 30 (10.8%) students used AI-based systems with a frequency of less than five times a week and only 9 (3.2%) respondents reported using AI-based systems for learning less than five times a week.

*4.2. Results of Measurement Model*

After conducting the descriptive analysis with SPSS 23.0, it was calculated that the skewness value of all items varied from −1.023 to −0.538 and the kurtosis values ranged from −0.693 to 0.832. Therefore, the collected data were verified to be evenly distributed since the absolute-value of skewness is lower than 3 and the absolute-value of kurtosis is lower than 8 [43,45]. Additionally, the factor loadings (λ value) of all items were examined with PLS-SEM and it was found they were all higher than 0.7, and all items were retained [43,46].

As suggested by prior studies, outer loadings of the items and average variance extracted (AVE) were measures to establish the convergent validity of the proposed research model [47,48]. The threshold value of factor loadings is equal to or greater than 0.7, and the value of AVE should exceed 0.5 to be accepted. Additionally, the Cronbach's alpha and composite reliability (CR) values should be equal to or greater than 0.7 as well.

As shown in Table 2, the standardized factor loadings of all items ranged from 0.794 to 0.891, which is higher than the recommended value of 0.7. Moreover, AVE was above the recommended value of 0.5, ranging from 0.682 to 0.724. Thus, the convergent validity of the measurement was confirmed [47]. Moreover, Cronbach's alpha varied from 0.883 to 0.924 and the values of CR ranged from 0.914 to 0.94, which all exceeded the threshold value of 0.7, demonstrating the internal consistency of the proposed research model.

**Table 2.** Results of measurement model.

| Variables | Items | Standardized Factor Loading | Cronbach's Alpha | CR | AVE |
|---|---|---|---|---|---|
| Perceived ease of use (PEOU) | PEOU1 | 0.829 | 0.883 | 0.914 | 0.682 |
| | PEOU2 | 0.794 | | | |
| | PEOU3 | 0.829 | | | |
| | PEOU4 | 0.840 | | | |
| | PEOU5 | 0.835 | | | |
| Perceived usefulness (PU) | PU1 | 0.863 | 0.912 | 0.935 | 0.741 |
| | PU2 | 0.876 | | | |
| | PU3 | 0.858 | | | |
| | PU4 | 0.854 | | | |
| | PU5 | 0.851 | | | |
| Attitude (Att) | Att1 | 0.881 | 0.901 | 0.931 | 0.770 |
| | Att2 | 0.868 | | | |
| | Att3 | 0.888 | | | |
| | Att4 | 0.873 | | | |
| Learning motivation-Learning interest (MLI) | MLI1 | 0.847 | 0.903 | 0.928 | 0.721 |
| | MLI2 | 0.848 | | | |
| | MLI3 | 0.849 | | | |
| | MLI4 | 0.848 | | | |
| | MLI5 | 0.855 | | | |

**Table 2.** *Cont.*

| Variables | Items | Standardized Factor Loading | Cronbach's Alpha | CR | AVE |
|---|---|---|---|---|---|
| Learning motivation-Achieving goals (MAG) | MAG1 | 0.869 | 0.889 | 0.923 | 0.750 |
| | MAG2 | 0.860 | | | |
| | MAG3 | 0.884 | | | |
| | MAG4 | 0.852 | | | |
| Learning motivation-Subjective norm (MSN) | MSN1 | 0.849 | 0.924 | 0.940 | 0.724 |
| | MSN2 | 0.834 | | | |
| | MSN3 | 0.866 | | | |
| | MSN4 | 0.844 | | | |
| | MSN5 | 0.854 | | | |
| | MSN6 | 0.858 | | | |
| Behavioral intention (BI) | BI1 | 0.853 | 0.884 | 0.920 | 0.741 |
| | BI2 | 0.868 | | | |
| | BI3 | 0.847 | | | |
| | BI4 | 0.876 | | | |
| Actual use (AU) | AU1 | 0.891 | 0.884 | 0.920 | 0.743 |
| | AU2 | 0.827 | | | |
| | AU3 | 0.862 | | | |
| | AU4 | 0.866 | | | |

It was suggested that there were two types of validities for evaluating the measurement model, namely convergent validity and discriminant validity [11,27]. Additionally, the convergent validity has been verified by the factor loadings and AVE of each construct and their indicators. As for the determination of the discriminant validity of PLS-SEM, three measures should be taken into consideration: the Fornell–Larcker criterion, the Heterotrait–Monotrait ratio of correlations (HTMT), and cross-loadings [47–51].

The first condition of discriminant validity is the Fornell–Lacker criterion, which suggests that the square root of AVE (diagonal value) in every construct in the correlation matrix should surpass the correlation coefficients of latent constructs. As indicated in Table 3, it could be observed that the discriminant validity of the present study has met the Fornell–Larcker criterion since the numbers in the diagonal are all higher than other numbers in the table. Furthermore, the second criterion of discriminant validity is HTMT, which is defined as the mean value of the item correlations across constructs (i.e., the heterotrait– heteromethod correlations) related to the (geometric) mean of the average correlations for the items measuring the same construct (i.e., the monotrait–heteromethod correlations). The values of HTMT must be less than 0.85 in order to confirm the discriminant validity [50]. According to Table 4, it is obvious that HTMT criterion has been met since all values are lower than 0.85, thus indicating the discriminant validity was further fulfilled. At last, the third condition of discriminant validity is cross-loadings, and the loading of each indicator should be higher than the loadings of its corresponding variables' indicators. As could be concluded from Table 5, the cross-loadings criterion has been fulfilled. Collectively, the discriminant validity among different constructs has been confirmed.

**Table 3.** Discriminant validity and the correlations of variables (Fornell–Larcker criterion).

| Variables | PEOU | PU | Att | MLI | MAG | MSN | BI | AU |
|---|---|---|---|---|---|---|---|---|
| PEOU | **0.826** | | | | | | | |
| PU | 0.706 | **0.861** | | | | | | |
| Att | 0.680 | 0.566 | **0.878** | | | | | |
| MLI | 0.635 | 0.532 | 0.546 | **0.849** | | | | |
| MAG | 0.687 | 0.554 | 0.502 | 0.537 | **0.866** | | | |
| MSN | 0.696 | 0.571 | 0.521 | 0.561 | 0.545 | **0.851** | | |
| BI | 0.699 | 0.614 | 0.586 | 0.548 | 0.556 | 0.593 | **0.861** | |

**Table 3.** *Cont.*

| Variables | PEOU | PU | Att | MLI | MAG | MSN | BI | AU |
|---|---|---|---|---|---|---|---|---|
| AU | 0.628 | 0.555 | 0.519 | 0.531 | 0.540 | 0.575 | 0.488 | **0.862** |

Note: The numbers in diagonal (in bold) are the square roots of AVE and other numbers are the correlation coefficients of each construct.

**Table 4.** Results of heterotrait–monotrait ratio (HTMT) discriminant validity.

| Variables | PEOU | PU | Att | MLI | MAG | MSN | BI | AU |
|---|---|---|---|---|---|---|---|---|
| PEOU | - | | | | | | | |
| PU | 0.784 | | | | | | | |
| Att | 0.759 | 0.622 | | | | | | |
| MLI | 0.709 | 0.583 | 0.603 | | | | | |
| MAG | 0.774 | 0.613 | 0.559 | 0.598 | | | | |
| MSN | 0.768 | 0.620 | 0.570 | 0.613 | 0.600 | | | |
| BI | 0.790 | 0.682 | 0.654 | 0.610 | 0.625 | 0.655 | | |
| AU | 0.709 | 0.618 | 0.580 | 0.592 | 0.609 | 0.634 | 0.549 | - |

**Table 5.** Results of cross-loadings.

| Variables | AU | Att | BI | MAG | MLI | MSN | PEOU | PU |
|---|---|---|---|---|---|---|---|---|
| ATT1 | 0.478 | **0.881** | 0.539 | 0.517 | 0.469 | 0.465 | 0.612 | 0.533 |
| ATT2 | 0.432 | **0.868** | 0.536 | 0.449 | 0.507 | 0.478 | 0.628 | 0.502 |
| ATT3 | 0.432 | **0.888** | 0.478 | 0.398 | 0.441 | 0.425 | 0.575 | 0.483 |
| ATT4 | 0.478 | **0.873** | 0.498 | 0.391 | 0.497 | 0.457 | 0.567 | 0.465 |
| BI1 | 0.403 | 0.476 | **0.853** | 0.484 | 0.436 | 0.505 | 0.605 | 0.521 |
| BI2 | 0.383 | 0.499 | **0.868** | 0.455 | 0.435 | 0.502 | 0.588 | 0.507 |
| BI3 | 0.432 | 0.506 | **0.847** | 0.484 | 0.475 | 0.501 | 0.593 | 0.521 |
| BI4 | 0.458 | 0.534 | **0.876** | 0.490 | 0.535 | 0.533 | 0.621 | 0.562 |
| MAG1 | 0.468 | 0.462 | 0.422 | **0.869** | 0.436 | 0.457 | 0.595 | 0.460 |
| MAG2 | 0.480 | 0.413 | 0.456 | **0.860** | 0.442 | 0.445 | 0.568 | 0.445 |
| MAG3 | 0.455 | 0.435 | 0.516 | **0.884** | 0.447 | 0.491 | 0.598 | 0.493 |
| MAG4 | 0.469 | 0.431 | 0.526 | **0.852** | 0.533 | 0.494 | 0.617 | 0.516 |
| MLI1 | 0.412 | 0.436 | 0.479 | 0.436 | **0.847** | 0.470 | 0.527 | 0.458 |
| MLI2 | 0.448 | 0.462 | 0.475 | 0.439 | **0.848** | 0.481 | 0.546 | 0.436 |
| MLI3 | 0.456 | 0.443 | 0.413 | 0.473 | **0.849** | 0.450 | 0.503 | 0.423 |
| MLI4 | 0.485 | 0.481 | 0.456 | 0.440 | **0.848** | 0.510 | 0.559 | 0.463 |
| MLI5 | 0.453 | 0.491 | 0.501 | 0.492 | **0.855** | 0.467 | 0.557 | 0.474 |
| MSN1 | 0.501 | 0.441 | 0.536 | 0.487 | 0.542 | **0.849** | 0.637 | 0.548 |
| MSN2 | 0.485 | 0.464 | 0.520 | 0.483 | 0.461 | **0.834** | 0.579 | 0.439 |
| MSN3 | 0.517 | 0.435 | 0.476 | 0.437 | 0.430 | **0.866** | 0.582 | 0.496 |
| MSN4 | 0.457 | 0.422 | 0.481 | 0.449 | 0.473 | **0.844** | 0.564 | 0.441 |
| MSN5 | 0.491 | 0.444 | 0.534 | 0.457 | 0.455 | **0.854** | 0.603 | 0.520 |
| MSN6 | 0.480 | 0.453 | 0.474 | 0.470 | 0.498 | **0.858** | 0.585 | 0.464 |
| Att1 | **0.891** | 0.499 | 0.455 | 0.488 | 0.516 | 0.535 | 0.593 | 0.509 |
| Att2 | **0.827** | 0.424 | 0.386 | 0.461 | 0.452 | 0.470 | 0.521 | 0.460 |
| Att3 | **0.862** | 0.401 | 0.417 | 0.473 | 0.419 | 0.487 | 0.518 | 0.479 |
| Att4 | **0.866** | 0.460 | 0.422 | 0.437 | 0.438 | 0.485 | 0.528 | 0.462 |
| PEOU1 | 0.503 | 0.524 | 0.578 | 0.571 | 0.520 | 0.532 | **0.829** | 0.546 |
| PEOU2 | 0.530 | 0.607 | 0.564 | 0.566 | 0.480 | 0.573 | **0.794** | 0.593 |
| PEOU3 | 0.532 | 0.576 | 0.612 | 0.582 | 0.552 | 0.646 | **0.829** | 0.629 |
| PEOU4 | 0.490 | 0.544 | 0.598 | 0.543 | 0.552 | 0.539 | **0.840** | 0.559 |
| PEOU5 | 0.533 | 0.550 | 0.533 | 0.572 | 0.514 | 0.576 | **0.835** | 0.580 |
| PU1 | 0.490 | 0.511 | 0.534 | 0.475 | 0.499 | 0.462 | 0.609 | **0.863** |
| PU2 | 0.477 | 0.512 | 0.552 | 0.495 | 0.503 | 0.506 | 0.634 | **0.876** |
| PU3 | 0.511 | 0.503 | 0.512 | 0.453 | 0.437 | 0.479 | 0.578 | **0.858** |
| PU4 | 0.483 | 0.425 | 0.513 | 0.479 | 0.398 | 0.495 | 0.608 | **0.854** |
| PU5 | 0.429 | 0.482 | 0.528 | 0.479 | 0.445 | 0.516 | 0.606 | **0.851** |

## 4.3. Structural Model Evaluation

By performing PLS algorithm and Bootstrap re-sampling method (5000), the research hypotheses have been examined. In the meantime, in order to detect the problem of multicollinearity of the structural model, the variance inflation factor (VIF) was calculated as well. As shown in Table 6, all values of VIF were lower than 5, indicating there is no significant multicollinearity among constructs. Additionally, the global goodness of fit index (GOD index) of the research model is 0.588, which is above the high standard of model fit 0.36 [52,53]. The explanatory power of the model is evaluated by measuring the discrepancy amount in the dependent variables of the model. R square and the path coefficients are essential measures for assessing the structural model [47,52]. The predictive accuracy of the model is determined using $R^2$ and the hypothesized association in the developed model was verified by path coefficients. Results related to R square, path coefficients, *t*-value, *p*-value, and VIF were synthesized in Table 6.

**Table 6.** Results of structural model and path coefficients.

| Criterion Variables | Predictor Variables | H | R Square | Path Coefficients | *t* Value | *p* Value | VIF | Results |
|---|---|---|---|---|---|---|---|---|
| PU | PEOU | H1 | 0.498 | 0.706 | 19.329 | 0.000 | 1.000 | Supported |
| Att | PEOU | H2 | 0.477 | 0.558 | 8.669 | 0.000 | 1.993 | Supported |
| | PU | H3 | | 0.172 | 2.257 | 0.012 | 1.993 | Supported |
| MLI | PEOU | H6a | 0.436 | 0.411 | 5.087 | 0.000 | 2.588 | Supported |
| | PU | H5a | | 0.133 | 1.766 | 0.039 | 2.050 | Supported |
| | Att | H7a | | 0.191 | 2.415 | 0.008 | 1.912 | Supported |
| MAG | PEOU | H6b | 0.482 | 0.565 | 6.882 | 0.000 | 2.588 | Supported |
| | PU | H5b | | 0.129 | 1.688 | 0.046 | 2.050 | Supported |
| | Att | H7b | | 0.045 | 0.596 | 0.276 | 1.912 | Rejected |
| MSN | PEOU | H6c | 0.500 | 0.548 | 6.926 | 0.000 | 2.588 | Supported |
| | PU | H5c | | 0.148 | 2.020 | 0.044 | 2.050 | Supported |
| | Att | H7c | | 0.065 | 0.952 | 0.170 | 1.912 | Rejected |
| BI | PU | H4 | 0.509 | 0.294 | 4.382 | 0.000 | 1.780 | Supported |
| | MLI | H8a | | 0.165 | 2.640 | 0.004 | 1.724 | Supported |
| | MAG | H8b | | 0.176 | 2.649 | 0.004 | 1.735 | Supported |
| | MSN | H8c | | 0.237 | 3.923 | 0.000 | 1.823 | Supported |
| AU | MLI | H9a | 0.436 | 0.203 | 3.056 | 0.001 | 1.738 | Supported |
| | MAG | H9b | | 0.230 | 2.874 | 0.002 | 1.723 | Supported |
| | MSN | H9c | | 0.290 | 4.216 | 0.000 | 1.858 | Supported |
| | BI | H10 | | 0.077 | 1.085 | 0.139 | 1.854 | Rejected |

As can be observed from Table 6, the $R^2$ values for the PU, Att, MLI, MAG, MSN, BI and AU ranged from 0.436 to 0.509, indicating the predictive power of these constructs is considered moderate because according to the recommendations provided by the study [49], when the $R^2$ value exceeds 0.67, it is perceived as "high", and the values between 0.33 and 0.67 are considered "moderate", while the values between 0.19 and 0.33 are treated as "weak". Overall, the $R^2$ value of the AU was found to explain 43.6% of the variance, suggesting the predictive power of these constructs was "moderate".

The one-tailed *t*-test entails that the 0.05 significance level ($p < 0.05$) requires a *t*-value > 1.657, the 0.01 significance ($p < 0.05$) requires a *t*-value > 2.354, and the 0.001 significance level ($p < 0.001$) requires a *t*-value > 3.152. The results of the proposed hypotheses were indicated in Table 6 and Figure 2. As can be seen, hypotheses regarding the impacts of Att on achieving goals of learning motivation ($\beta = 0.045$, $p > 0.05$), and subjective norm of learning motivation ($\beta = 0.065$, $p > 0.05$), and BI on AU were rejected by the empirical investigation (H7b, H7c and H10, respectively). However, PEOU of AI-based systems was proven to affect PU significantly and positively ($\beta = 0.706$, $p < 0.000$), students' Att

($\beta = 0.558$, $p < 0.000$), learning interest in learning motivation (($\beta = 0.411$, $p < 0.000$), achievement of goals of learning motivation (($\beta = 0.565$, $p < 0.000$) and subjective norm of learning motivation ($\beta = 0.548$, $p < 0.000$), thus supporting H1, H2, H6a, H6b, H6c. Furthermore, PU of AI-based systems was found to positively influence students' Att ($\beta = 0.172$, $p < 0.05$), learning interest ($\beta = 0.133$, $p < 0.05$), achievement of goals ($\beta = 0.129$, $p < 0.05$), subjective norm ($\beta = 0.148$, $p < 0.05$) of learning motivation and students' BI ($\beta = 0.294$, $p < 0.001$); hence, the hypotheses H3, H5a, H5b, H5c and H4 have been confirmed.

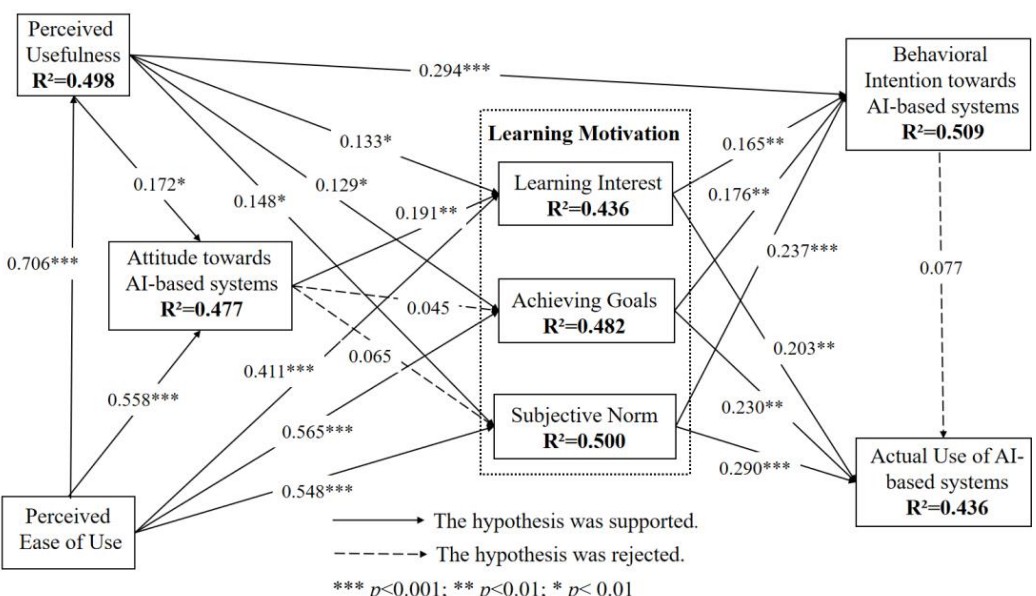

**Figure 2.** Results of the structural model.

In the meantime, students' Att was proven to significantly affect students' learning interest among three types of learning motivation only ($\beta = 0.191$, $p < 0.01$), supported by H7a. As for the association between students' learning motivation and their BI, hypotheses H8a (MLI→I, $\beta = 0.165$, $p < 0.01$), H8b (MAG→BI, $\beta = 0.176$, $p < 0.01$) and H8c (MSN→BI, $\beta = 0.237$, $p < 0.000$) were confirmed. For the relation between students' learning motivation and their AU of AI-based systems, hypotheses H9a (MLI→AU, $\beta = 0.203$, $p < 0.01$), H9b (MAG→AU, $\beta = 0.230$, $p < 0.01$) and H9c (MSN→AU, $\beta = 0.290$, $p < 0.000$) were confirmed as well.

## 5. Conclusions and Discussions

### 5.1. Conclusions

With the great popularity and potential of AI-based systems or programs in the context of education, the present research was performed to identify determinants contributing to the AU of AI-based systems among college students. After conducting empirical investigation, it was proven that college students' AU of AI-based systems depends on a combination of college students' PU, PEOU, Att towards AI-based systems and different dimensions of their learning motivation.

Specifically, students' PEOU towards AI-based systems significantly and positively impacts PU (supported by H1), and this finding is very much in line with the results verified by other studies, which illustrated that in educational settings such as e-learning, K-MOOC, Zoom, etc., PEOU does have influence on PU [12,14,54]. Furthermore, PEOU and PU were reported to significantly affect students' Att towards AI-based systems (supported by H2 and H3, respectively), and PU positively and directly affect students' BI (supported by H4), which is consistent with the basic content of TAM regarding the relationship among PEOU, PU, and users' Att and BI towards certain technology, program, or system [14,55,56]. In terms of students' learning motivation, the empirical results indicated that PEOU and PU

directly affect three aspects of students' learning motivation (supported by H5 and H6), while the relationship among Att and students' learning motivation of achieving goals and subjective norm was reported to be statistically insignificant (rejected by H7b and H7c). This result disagrees with the content of attitude/motivation test battery in which learners' Att could positively impact on their motivation [31]. This discrepancy observed in the current study might refer to the setting in which the application of attitude/motivation test battery is specific to foreign language learning. However, the setting for the present study refers to the application of AI-based systems for students' diverse learning purposes, such as programming, simulation exercise by VR, personal learning evaluation, etc. Additionally, this has been consistent with the study that AI could help education in terms of educational ambit and content regarding what kind of education is needed [57].

As for the influences of learning motivation on students' BI and AU of AI-based systems, they were verified by H8 and H9, respectively. Among three dimensions of students' learning motivation, subjective norm was examined to be the most significant factor contributing to students' BI and their AU. Additionally, this result could be the data source for AI-based systems' expansion among students, for example, developers or operators of AI-based systems should pay more attention to arouse the attention of and obtain support from teachers, parents, or influential educators. Additionally, this has been examined in relation to Azerbaijan students' BI to use e-learning, in which subjective norm has a positive and significant impact on BI to use e-learning [58].

*5.2. Implications*

The findings of the present study could generate some implications for the application of AI-based systems among college students. Firstly, this study could be a starting point for future research regarding technology acceptance of AI-based systems in the educational setting, especially for higher education. The casual relationships among students' PEOU, PU, Att, learning motivation, BI and AU of AI-based systems have been examined and verified. Students' learning motivation, as a mediator influencing students' BI and AU towards AI-based systems, should attract scholars' attention for future research. Secondly, the significant correlation among students' learning motivation, BI and AU towards AI-based systems suggested that for the expansion and effective application of AI-based systems among students, it is instrumental and essential to arouse students' interest and help them build specific goals. More importantly, the endorsement from professionals is vital as well. Third, the mediating effects of Att on the path from PU to learning motivation and PEOU to learning motivation were empirically rejected for the dimensions of achieving goals and subjective norm. However, these were found to positively affect students' learning interest. As such, for future research regarding the application of AI-based systems to motivate students' learning behavior, the enhancement of students' learning interest could be one of the focusing points. Fourth, the denied relationship between students' BI and their AU suggested that the mediating role of BI in students' learning motivation and AU was insignificant. This finding could provide baseline data for future studies investigating the relationship among students' learning motivation, BI and AU towards specific technology, system, or program. Lastly, the empirical findings of this research could provide stakeholders, systems developers or operators with statistic information to establish effective decisions related to the acceptance of AI-based systems in the context of higher education and other similar contexts.

*5.3. Discussions*

Though the study has been completed, several limitations should be taken into consideration. Firstly, the AU of the present study was applied subjective measures to evaluate students' actual usage of AI-based systems. However, according to the prior studies, it is important to measure AU objectively as there is a difference in the relationship between the TAM variables, subjective and objective measures of actual technology use [39,59,60]. Therefore, for future research, if it is possible, it will be better to adopt objective mea-

sures to evaluate users' actual technology/systems/programs usage such as the number of log-ons or the number of hits-on of targeted technology or program. Secondly, based on the IP address provided by Wenjuanxing.com, it could be observed that most survey respondents are from the same province in China mainland; therefore, the discrepancy caused by different areas, such as educational standard, level of economic development and so forth, was ignored. As such, the application of research findings in other areas should be encouraged with caution. Additionally, for future research, it would be more conclusive to conduct the survey with respondents from different areas. Thirdly, students' socio-demographic characteristics, such as using frequency, gender, specialization, etc., which could be important factors contributing to their subsequent BI and AU, were not investigated in the proposed conceptual model. Hence, for future research, it could be better to investigate the moderating effects of these variables. Finally, students responded that they have used AI-based systems for different learning purposes. Hence, it would be more accurate to understand students' behavior by further analyzing the influence of their learning purposes.

**Funding:** This research was financially supported by Zhejiang Province Education Science Planning Project (No. 2022DJG013) and Soft Science Research Base of Water Digital Economy and Sustainable Development Research of Zhejiang University of Water Resources and Electric Power.

**Institutional Review Board Statement:** Not applicable.

**Informed Consent Statement:** Not applicable.

**Data Availability Statement:** Not applicable.

**Conflicts of Interest:** The author declares no conflict of interest.

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
