# Peer review of "Determinants of College Students’ Actual Use of AI-Based Systems: An Extension of the Technology Acceptance Model"

_sustainability, doi:10.3390/su15065221_

Round 1

Reviewer 1 Report

Although the idea underlying this piece of research is very interesting, I believe the paper needs to be amended due to a serious of factors: 

1.   What type of AI-based systems do the authors refer to in this research? This should be further clarified in the introduction or Lit. Rev. sections.  There are some references such as ‘Specifically, it was suggested by the study that the application of AI-based chatbots in the review process of public health courses’ (101) but it remains unclear what ‘type/s of AI’ are the object of study (context, research participants).

2. Data collection. Further details should be provided about research participants and previous experience with AI-based types.

3. More details about the AI-based systems used in this research are necessary. The using frequency factor is not well explained.  ‘In terms of the using frequency, the majority of respondents (168) used AI-based systems for learning at least once per day with the percentage of 58.4%, followed by twice to five times a week 237 (72, 25.8%). 30 (10.8%) students used AI-based systems with the frequency less than five times a week and only 9 (3.2%) respondents showed using AI-based systems for learning less than five times a week. (235-243)’.   And how could the frequency of using AI-based systems affect the results?

4. Table number is missing in ‘As indicated in Table…, it could see’ (268)

5. Rephrase (to be significantly impact on): ‘Furthermore, PEOU and PU were examined to be significantly impact on students’ Att towards AI-based systems’ (340)

6. Conclusions:  Some statements need to be better supported. For example ‘However, the setting for the present study refers to the application of AI-based systems for students’ diverse learning purposes’ (353).  The authors should also refer to recent studies which demonstrated that the TAM results for each dimension regarding the use of AI-based systems may depend on context, setting and area of study, for example:

*Alam, A. (2021, November). Possibilities and apprehensions in the landscape of artificial intelligence in education. In 2021 International Conference on Computational Intelligence and Computing Applications (ICCICA) (pp. 1-8). IEEE.

*Belda-Medina, J., & Calvo-Ferrer, J. R. (2022). Using Chatbots as AI Conversational Partners in Language Learning. Applied Sciences, 12(17), 8427.

7. Some statements need further clarification, for example ‘As such, for the future research regarding the application of AI-based systems, to some extent, the function of users’ Att could be neglected.’ (375-377) Why? Attitudes can be a key factor. 

8. Since this research is based on ‘the application of AI-based systems for students’ diverse learning purposes.’ (353), did the authors observe any difference among participants depending on the discipline/learning purposes? Did they perform any analysis?

Reviewer 2 Report

Add some recent previous literature studies on the evaluation of learning using AI, especially in terms of TAM, it would be better! to see the novelty of this research

-------

From all the respondent data obtained, was there a mechanism to verify and validate whether the respondent has actually used AI in learning? If so, please describe! and were there any invalid respondent data after going through the process?

Reviewer 3 Report

1. A huge number of hypotheses. What for? They need to be enlarged, and those that have already been verified in previous studies must be removed. It turns out that all is bound up with everything else.

2. The survey was administered to college students. Since birth this generation representatives have been actively engaged in activities driven by modern technologies and to use all the innovations is not a problem for them. In this regard, the study into this topic with the involvement of older age groups is likely to be more re

3. A greater variety of groups under consideration (with breaking into males, females, students majoring in different sciences, those who are academically successful and those who are not, etc.) can improve the text significantly.

4. What information do tables 3, 4 и 5 provide? These results can be described simply.

5.  There is a lack of results and conclusions.

6.  The description of similar studies is missing.

Round 2

Reviewer 1 Report

Thanks for the revised version.

Reviewer 3 Report

The authors' answers to the questions raised are quite satisfactory.